# Optimization of Ultrasonic-Assisted Extraction of Total Flavonoids from Abrus Cantoniensis (*Abriherba*) by Response Surface Methodology and Evaluation of Its Anti-Inflammatory Effect

**DOI:** 10.3390/molecules27072036

**Published:** 2022-03-22

**Authors:** En-Yun Wu, Wen-Jing Sun, Ying Wang, Ge-Yin Zhang, Bai-Chang Xu, Xiao-Gang Chen, Kai-Yuan Hao, Ling-Zhi He, Hong-Bin Si

**Affiliations:** 1State Key Laboratory for Conservation and Utilization of Subtropical Agro-Bioresources, College of Animal Science and Technology, Guangxi University, Nanning 530004, China; 1918302031@st.gxu.edu.cn (E.-Y.W.); 1918393051@st.gxu.edu.cn (Y.W.); 2118402012@st.gxu.edu.cn (G.-Y.Z.); 1918393057@st.gxu.edu.cn (B.-C.X.); 1918393006@st.gxu.edu.cn (X.-G.C.); 1918302008@st.gxu.edu.cn (K.-Y.H.); 2018393020@st.gxu.edu.cn (L.-Z.H.); 2Guangxi Key Laboratory of Agricultural Resources Chemistry and Biotechnology, College of Biology & Pharmacy, Yulin Normal University, No. 1303 Jiaoyu East Road, Yulin 537000, China; wenjingsun@gxu.edu.cn

**Keywords:** Abrus cantoniensis, flavonoids, response surface, ultrasonic extraction, anti-inflammatory

## Abstract

Abrus cantoniensis is a Chinese herbal medicine with efficacy in clearing heat and detoxification, as well as relieving liver pain. The whole plant, except the seeds, can be used and consumed. Flavonoids have been found in modern pharmacological studies to have important biological activities, such as anti-inflammatory, antibacterial and antioxidant properties. The antibacterial and antioxidant bioactivities of the total flavonoids of Abrus cantoniensis (ATF) have been widely reported in national and international journals, but there are fewer studies on their anti-inflammatory effects. The present study focused on the optimization of the ultrasonic extraction process of ATF by response surface methodology and the study of its anti-inflammatory effects in vitro and in vivo. The results showed that the factors that had a great impact on the ATF extraction were the material-to-liquid ratio, ultrasonic extraction cycles and ethanol concentration. The best extraction process used a material-to-liquid ratio of 1:47, ultrasonic extraction cycles of 4 times, an ethanol concentration of 50%, an ultrasonic extraction time of 40 min and an ultrasonic power of 125 W. Under these conditions, the actual extraction rate of total flavonoids was 3.68%, which was not significantly different from the predicted value of 3.71%. In an in vitro anti-inflammatory assay, ATF was found to be effective in alleviating LPS (lipopolysaccharide)-induced inflammation in mouse peritoneal macrophages. In an in vivo anti-inflammatory assay, ATF was found to have a significant inhibitory effect on xylene-induced ear swelling in mice and cotton ball granuloma in mice, and the inhibitory effect was close to that of the positive control drug dexamethasone. This may provide a theoretical basis for the further development of the medicinal value of Abrus cantoniensis.

## 1. Introduction

Abrus cantoniensis is a member of the legume family Acaciaceae and is also known as magnoflorine, red hen’s wort, yellow food grass, and clover lepidotrichia. Apart from the seeds, the whole plant can be used as medicine and food; this occurs mainly in Guangxi, Guangdong, Hunan and other provinces in China. The plant demonstrates the effects of clearing heat, detoxifying the liver and relieving pain [1,2,3]. Studies have shown that the whole plant of chicken bone and grass is rich in chemical components, mainly triterpenoids, flavonoids, polysaccharides, alkaloids, anthraquinones, and other components which are abundant in medicinal value [4]. There are many ATF extraction methods, such as extraction, ultrafiltration, microwave digestion, and ultrasonic extraction. Ultrasonic-assisted solvent extraction of plant secondary metabolites has been widely used in the fields of medicine, food and chemistry. It is a green extraction technique with a high extraction rate, low energy consumption and a short extraction cycle [5,6,7]. Spanish food engineering researchers used traditional solvent and ultrasound-assisted extraction methods to extract flavonoids from grapefruit solid waste [8], and found that ultrasound-assisted extraction significantly increased the extraction rate of flavonoids at low ethanol levels, low temperatures and shorter time periods, while increasing the antioxidant activity of the flavonoids by 76%, achieving a more environmentally friendly and less time-consuming method than traditional solvents. In previous studies, the content of total flavonoids was determined, which proved that Abrus cantoniensis was rich in flavonoids [9]. One study used orthogonal tests to optimize the ultrasonic extraction method and the ethanol reflux method for the total flavonoids, and found that the extraction rate of total flavonoids by the ethanol reflux method was slightly higher than the ultrasonic method, but the time consumed for ultrasonic extraction was significantly less than the ethanol reflux method [10]. In addition, a single factor test and orthogonal test were used to optimize the ATF ultrasonic extraction process, and it was found that the factors that mainly affected the ultrasonic extraction were the material to liquid ratio, ultrasonic extraction cycles, ethanol concentration, and ultrasonic time. It was also found that the ultrasonic extraction solution had a certain scavenging effect on hydroxyl radicals and superoxide ion radicals [11].

The response surface method (RSM) has become a common optimization method in recent years. Under the premise of reasonable design, this method establishes a binary regression equation model by using a certain amount of test data. On the basis of guaranteeing the accuracy and reliability of the model, it reasonably predicts a combination of advantages in order to obtain the best test parameters [12]. In the process of response surface method optimization, all levels of test factors are analyzed, which overcomes the limitation of previous orthogonal experiments that could only analyze isolated points, but could not give intuitive graphics [13].

In modern pharmacological studies, it has been shown that Abrus cantoniensis has hepatoprotective, choleretic, antibacterial, anti-inflammatory, immune-enhancing, free-radical-scavenging, smooth-muscle-function-regulating and endurance-enhancing effects [3]. Studies have found phenolics and flavonoids with antioxidant effects in Abrus cantoniensis and Abrus mollis Hance, which can effectively prevent diseases related to oxidants and free radicals [4]. In a study comparing the anti-inflammatory and immune effects of Abrus cantoniensis and Abrus mollis Hance, it was found that both Abrus cantoniensis and Abrus mollis Hance have some anti-inflammatory potency [14]. The total flavonoids and L-abrine of Abrus cantoniensis were both found to have anti-inflammatory effects [15]. In order to further improve the ATF ultrasonic extraction rate, this paper used the response surface methodology to optimize the ATF ultrasonic extraction based on the results of the single-factor test, and the anti-inflammatory effect of ATF was studied based on in vitro and in vivo anti-inflammatory assays, which can provide a theoretical basis for the further development and utilization of Abrus cantoniensis.

## 2. Results and Discussion

### 2.1. Single Factor Experiment Results

#### 2.1.1. Influence of Material-to-Liquid Ratio

Using the single-factor method, a reasonable range of five factors (material-to-liquid ratio, ultrasonic extraction cycles, ethanol concentration, ultrasonic time and ultrasonic power) affecting the ATF extraction rate was screened out. The results are shown in Figure 1. Under the constant conditions of an ethanol concentration of 70%, an ultrasonic power of 150 W, an ultrasonic time of 15 min, and a number of ultrasonic cycles of 3 times, the extraction rates of ATF measured by varying the material-liquid ratio of 1:20, 1:30, 1:40, 1:50 and 1:60 are shown in Figure 1a. The results showed that the extraction rate of ATF gradually increased with increasing material-liquid ratio, and began to have a decreasing trend when the material-liquid ratio was greater than 1:50. The reason may be that if the material-liquid ratio is too small, the corresponding extractant is too small, so that ATF cannot be fully dissolved in the solvent; if the material-liquid ratio is too large, other substances may be dissolved, such as polysaccharides [16], resulting in a decrease in the extraction rate of ATF. Therefore, the best extraction material-liquid ratio was shown to be 1:50.

#### 2.1.2. Influence of Ultrasonic Extraction Cycles

Under the constant conditions of a material-to-liquid ratio of 1:50, ethanol concentration of 70%, sonication power rate of 250 W and sonication time of 15 min, the extraction rate of ATF measured by varying the sonication times of 2, 3, 4 and 5 times is shown in Figure 1b. The results showed that the extraction rate of ATF gradually increased with increasing sonication times, and began to have a decreasing trend when the sonication times were greater than three times. The reason may be that the number of ultrasound-assisted extractions is too small and the cell fragmentation is not complete [17], so that ATF are not all dissolved in the solvent; if the number of ultrasound extractions is too high, more impurities may be extracted, which may destroy the structure of the total flavonoids due to the thermal and mechanical effects brought about by ultrasound, which affects the extraction of ATF. Therefore, the optimal number of ultrasonic extractions was shown to be three times.

#### 2.1.3. Influence of Ethanol Concentration

Under the constant conditions of a material-liquid ratio of 1:50, an ultrasonic power of 150 W, an ultrasonic time of 15 min, ultrasonic extraction cycles of 3 times, and varying the ethanol concentration to 30%, 40%, 50%, 60%, 70%, 80% and 90%, the measured extraction rate of ATF is shown in Figure 1c. The results showed that the extraction yield of ATF increased gradually with increasing ethanol concentration, and a decreasing trend began when the ethanol concentration was greater than 40%. This may be because ethanol has better solubility and strong cell penetration; thus, the higher the concentration of ethanol, the more beneficial to the solubilization of flavonoids, but when ethanol exceeds a certain concentration, there will be an increase in some alcohol-soluble pigments and the leaching of strong lipophilic components [18]. These components will affect the solubilization of ATF such that the extraction rate of ATF decreases. Therefore, the optimal ethanol concentration for ultrasonic extraction was shown to be 40%.

#### 2.1.4. Influence of Ultrasonic Time

Under the constant conditions of a material-to-liquid ratio of 1:50, 70% ethanol concentration, 150 W ultrasonic power, 3 times sonication, etc., and varying the ultrasonic time as 10, 20, 30, 40 and 50 min, the measured extraction rate of ATF is shown in Figure 1d. The results showed that the extraction rate of ATF gradually increased with increasing ultrasonic time and began to have a decreasing trend when the ultrasonic time was greater than 40 min. The reason may be that with increased ultrasonic extraction time, the degree of cell fragmentation will also increase [19], and the extraction rate will have a substantial increase. However, with too long of an ultrasonic time, the extraction solution becomes turbid, and the viscosity will increase. In this case, the thermal and mechanical effects generated by the ultrasound may damage the ATF structure, affecting the extraction and determination of contents. Therefore, the optimal ultrasonic extraction time was selected to be 40 min.

#### 2.1.5. Influence of Ultrasonic Power

The ATF extraction yields obtained by varying the ultrasonic power (100 W, 125 W, 150 W, 175 W, 200 W, 225 W, 250 W) under the constant conditions of a material-to-liquid ratio of 1:50, 70% ethanol concentration, 15 min ultrasonic time and 3 times ultrasonic extraction cycles are shown in Figure 1e. The results showed that the ATF extraction yield increased gradually with increased ultrasonic power and began to have a decreasing trend when the ultrasonic power was greater than 150 W. This may be because the higher the ultrasonic power, the easier it is to break the cells and release the flavonoids in the solvent [20]. However, when the ultrasonic power is too high, it will have an effect on the stability of the flavonoids and change their structure [21], and the extraction rate of ATF will be reduced as a result. Therefore, the optimal ultrasonic power was shown to be 150 W.

### 2.2. Plackett-Burman Design (PBD) Experimental Results

The results of the PBD experiments are shown in Table 1 and Table 2 and Figure 2. In this experiment, five variables affecting the ATF extraction were evaluated using PBD in Design-Expert software, including the material-to-liquid ratio, number of ultrasound cycles, ethanol concentration, ultrasound time and ultrasound power. The results showed that the material-liquid ratio had the most significant effect, followed by the ethanol concentration. The weaker effect was on the number of sonication times, while sonication time and sonication frequency had no significant effect. The equation obtained from the PBD experiment is as follows:(1)$$\mathrm{Content}\left(\mathrm{ATF}\right)=3.70917-0.035917A+0.025083B-0.029167C$$
where *A* represents the material to liquid ratio (g/mL), *B* represents the ethanol concentration (%), and *C* represents the number of ultrasound cycles (time).

The results of the ANOVA showed in Table 2 that the obtained model had an F-value of 13.69 and a prob (P) > F-value of 0.0016, which was significant, indicating that the model fitted well throughout the regression area under study. The complex correlation coefficient R2 value of 0.8396 indicates that the three factors in the model, the material-to-liquid ratio, ethanol concentration and number of ultrasound cycles, are well correlated; the corrected coefficient of determination Adj R2 of 0.7758 is in reasonable agreement with the Pred R2 value of 0.6331. The smaller the C.V. value, the higher the accuracy and credibility of the experiment. The C.V. value in this experiment is 8.93%, indicating that the experiment has good accuracy and credibility [22]. The precision Adeq Precision value is 9.329, a value greater than 4. The result indicates that the effective signal-to-noise ratio of this experiment is reasonable. The results indicate that the effective signal-to-noise ratio of this experiment is reasonable. It is very necessary to check whether the fitted model can have an adequate approximation to the real system, and usually the residuals of the least square method are applied to judge the adequacy of the model [23]. The actual system was checked for normality assumptions by constructing a normal probability plot of the residuals (Figure 2a). To satisfy the assumption of normality, the residual plot needs to be approximately distributed along a straight line [24]. The results illustrate the plots of residuals vs. the predicted responses in Figure 2b. The distribution of the residuals is usually randomly dispersed, which means that the variance of the original observations of these graphs is constant for Y values [25]. From the above results, it can be seen that the empirical model can better represent the effects of three factors, namely the material-liquid ratio, ethanol concentration and the number of ultrasonic cycles.

### 2.3. Box-Behnken Design (BBD) Experimental Results

The results of the PBD experiments revealed that the material-liquid ratio, ethanol concentration and the number of ultrasound cycles had important effects on the ATF extraction. A BBD experimental design was carried out for these three factors, and the results are shown in Table 3. The following Equation (2) was obtained from the results of the BBD experiment:(2)$$\mathrm{Content}\left(\mathrm{ATF}\right)=2.45-0.14A+0.21B+0.27C-0.053AB+0.050AC+0.090BC-0.28A^{2}+0.41B^{2}+0.26C^{2}$$
where *A* represents the material to liquid ratio (g/mL), *B* represents the ethanol concentration (%), and *C* represents the ultrasonic extraction cycles (time).

The ANOVA results of the BBD experiment are presented in Table 4, and the results showed that the model had an F value of 21.75, a complex correlation coefficient R2 value of 0.9655, and a prob (P) > F value of 0.0003, which was highly significant. The three factors affecting the ultrasonic extraction had a better correlation, and the model fitted well in the whole regression area under study and could be used to predict the response values. In the results of the significance test of the coefficients of the quadratic model regression equation, the Prob (P) > F values of A, B and C were all less than 0.01, indicating a highly significant linear effect on the ATF ultrasonic extraction rate. The prob (P) > F values of A2, B2 and C2 were all less than 0.05, indicating that they had a highly significant curvilinear effect on the ultrasonic extraction rate. The prob (P) > F values of AB, AC and BC were all greater than 0.05, indicating that their interaction effects on the ultrasonic extraction rate were not significant. The response surface plots and their contour plots based on the multiple regression equations from the BBD experiments can be seen in Figure 3, which can be used to analyse and evaluate the interaction of any two factors influencing the ATF sonication rate in order to determine the range of values of the factors influencing the optimal sonication rate. In the response surface plots, the steeper the slope of the response surface, the more significant the effect of this factor on the response value, and the more sensitive the ATF extraction rate is to the change of this factor. The contour plot is consistent with the results of the corresponding surface plots; the closer the curve is to the centre, the larger the corresponding response value, and the closer its shape is to a circle, indicating a weaker interaction between the two independent variables. If it is closer to an ellipse, it indicates that the closer the curve is to the centre, the larger the corresponding response value is [24]. The results indicated that the material-to-liquid ratio, the number of times of ultrasonication and the concentration of ethanol had a significant effect on the ATF extraction, and their effects on the response values were: ethanol concentration > number of times of ultrasonication > material-to-liquid ratio. However, the interactions between the material-to-liquid ratio and the number of times of ultrasonication, material-to-liquid ratio and ethanol concentration, and the number of times of ultrasonication and ethanol concentration were not significant.

Based on the response surface results, the optimum process conditions for the extraction of ATF were a material-to-liquid ratio of 1:47.43, a number of sonications of 4 times and a concentration of ethanol of 50%. The predicted extraction rate under these conditions was 3.71%. Due to the practical operation of the experiment and the limitations of the apparatus itself, the optimal extraction process conditions were adjusted to the following values: the value of the material-to-liquid ratio was 1:47, the number of sonications was 4 times and the concentration of ethanol was 50%. The ATF extraction rate measured under the actual conditions was 3.68%, and the relative error between the predicted and actual values was calculated to be 0.81% (*n* = 3), indicating that the model fitted well with the actual situation. The model has some reliability.

### 2.4. Anti-Inflammatory Test Results

#### 2.4.1. ATF Effects on LPS-Induced Inflammation in Mouse Peritoneal Macrophages

##### Safe Concentrations of ATF and LPS on Mouse Peritoneal Macrophages

A prerequisite for the assessment of plant extracts is to ensure that their toxic effects are influenced by cellular, metabolic or signaling pathways that inhibit the production of pro-inflammatory factors and thus exert anti-inflammatory effects. Therefore, tests at non-toxic doses of drugs are the basis for examining the efficacy of drugs [26]. The safe concentrations of ATF and LPS on mouse peritoneal macrophages were determined by the MTT((3-(4,5-dimethylthiazol-2-yl)-2,5-diphenyltetrazolium bromide)) method, and the results are shown in Figure 3. Compared with the blank control (CG) group, ATF at concentrations of 100 μg/mL and 50 μg/mL had a significant proliferation-promoting effect on mouse peritoneal macrophages in Figure 4a. At a concentration of 12.5 μg/mL, the growth of mouse peritoneal macrophages was significantly inhibited. There was no inhibitory effect on the growth of mouse peritoneal macrophages at concentrations below 10 μg/mL. Taking into account that the anti-inflammatory properties of the organism are closely related to the increase in macrophages, the ATF concentrations chosen to perform the anti-inflammatory assay were 100 μg/mL, 50 μg/mL and 10 μg/mL. These results are consistent with the effect of an aqueous extract of pomegranate flower on mouse peritoneal macrophages [27]. It was suggested that the significant increase in absorbance may be due to the fact that this concentration stimulates the secretion of large amounts of succinate dehydrogenase (SDH) from macrophages, resulting in no change in quantity but increased activity and secretion of large amounts of SDH [27]. In Figure 4b, concentrations of LPS in the range of 1 to 100 μg/mL had no significant inhibitory effect on the growth of mouse peritoneal macrophages compared to the CG group, and the stimulation concentration of LPS in the relevant literature was mostly 1 μg/mL [28]. The concentration of LPS chosen to carry out the anti-inflammatory assay was 1 μg/mL, taking into account the literature and the economic benefits.

##### ATF Effects on NO Release

The results of the test using the nitric oxide (NO) test kit (nitrate reductase method) are shown in Figure 5a. NO levels were significantly higher in the LPS inflammation model control (MC) group compared to the no-treatment control (NT) group (*p* < 0. 05). Compared with the MC group, ATF significantly reduced the NO content of mouse peritoneal macrophages (*p* < 0. 05), and the ATF-H (1000 mg/kg) group had the best inhibitory effect. As a versatile mediator involved in a large number of pathological and physiological processes, NO is a pro-inflammatory molecule that plays an important role in the inflammatory response [29], and the amount of cellular NO secreted indirectly reflects the degree of inflammation that occurs [30]. NO is essential for macrophages to perform phagocytosis. As NO increases, macrophages have a stronger phagocytic function [31]. A study has shown that both 35% and 75% ethanol-eluting salvianol flavonoids significantly reduced the secretion of NO from LPS-stimulated macrophages in a dose-dependent manner [32]. At a concentration of 50 μg/mL of Echinacea adventitious root extract, the amount of NO released was significantly lower than in the LPS group, suggesting that Echinacea adventitious root also has anti-inflammatory effects [33]. This was consistent with the results of the present study, which showed that ATF significantly reduced the secretion of NO from macrophages stimulated by LPS and slowed down LPS-induced inflammation in mouse macrophages.

##### ATF Effects on the Release of Inflammatory Factors

The secretion of inflammatory factors such as IL-6, TNF-α, IL-1β and IL-10 was measured by ELISA kits, and the results are shown in Figure 5b–e. LPS-induced inflammation in mouse peritoneal macrophages is a common model of inflammation used in in vitro anti-inflammatory studies [34]. LPS induces the production of various inflammatory factors such as TNF-α, IL-1β and IL-6 [28]. Cytokines such as TNF-α and IL-6 have been reported to trigger inflammation by activating key proteins in signaling pathways, such as NF-KB and MAPK [35]. The production of pro-inflammatory factors IL-6, TNF-α and IL-1β was significantly higher in the MC group than in the NT group, which is consistent with the findings of Jain et al. [36]. The secretion of pro-inflammatory factors IL-6, TNF-α and IL-1β was significantly decreased in the ATF-L, ATF-M and ATF-H groups compared with the MC group, and there was no significant difference between the high-dose group of IL-6 and TNF-α with the NT group. For the anti-inflammatory factor IL-10, ATF significantly promoted its secretion, but there was a significant difference between the ATF-H group and the NT group. In the study of the anti-inflammatory effects of chemo flavonoids, it was found that the flavonoids could modulate the secretion of IL-10 by cells to inhibit the amplification of inflammatory responses [37], which is consistent with the results of the present study. Based on the above results, it is clear that ATF can effectively alleviate the inflammation of LPS on mouse peritoneal macrophages.

#### 2.4.2. Inhibition of Xylene-Induced Ear Swelling in Mice

Xylene can promote the release of vasoactive amines, such as mast cells in tissues, causing vasodilation and increased permeability in the ear of mice, leading to an inflammatory infiltrative response [38]. The inhibitory effect of ATF on xylene-induced ear swelling in mice is shown in Figure 6a. Compared with the no treatment model (NT) group, ATF had a significant inhibitory effect on ear swelling in mice caused by xylene, and the inhibitory effect was dose-dependent. The inhibition rate of dexamethasone, the positive control drug, on ear swelling in mice caused by xylene was 58.72%, while the inhibition rates of ATF-L, ATF-M and ATF-H on ear swelling in mice caused by xylene were 27.91%, 48.84% and 78.49%, respectively. The inhibition effect of the ATF-H group was comparable to that of the positive drug dexamethasone. In a previous study on the anti-inflammatory effect of Abrus cantoniensis [15], it was found that the maximum inhibition of ear swelling in mice was 76.17%, which was less than that of the high dose group in the experiment, probably because the total flavonoid content in the raw herb did not reach 1000 mg/kg, or the presence of non-flavonoid substances in Abrus cantoniensis could affect the anti-inflammatory effect of ATF.

#### 2.4.3. Inhibition of Cotton Pellet Granulomas in Mice

The results are shown in Figure 6b. ATF had a significant inhibitory effect on cotton ball granuloma in mice. The inhibition rate was 37.09% in the dexamethasone (5 mg/kg) treatment (DEX) group and 37.75%, 51.66% and 55.63% in the ATF-L, ATF-M and ATF-H groups, respectively. Compared with the positive control drug dexamethasone, there was no significant difference in the inhibitory effect of the ATF-L group, and the inhibitory effect of the ATF-M and ATF-H group was significantly better than that of the DEX group. Regarding the differences compared to previous studies in which ATF was negatively correlated with anti-inflammatory effects on chronic inflammation [15], there are some reasons for speculation. Firstly, geographic differences may have contributed, as the dry whole plant of Abrus cantoniensis was used in the study conducted in Liujiang of Liuzhou and Ruyang of Henan Province, and the dry whole strain of Abrus cantoniensis was used in this trial in Nanning, Guangxi, China. Secondly, differences in content and purity may have contributed; mice were treated by gavage in that study, while using the drug for distilled water decocted and concentrated to a certain volume of crude extract. In the present study, ATF underwent ultrasound assisted alcohol extraction after purification by D101 macroporous resin, and the obtained powder was freeze-dried and dissolved by a certain amount of distilled water.

## 3. Materials and Methods

### 3.1. Materials and Chemicals

The dried whole plant of Abrus cantoniensis was purchased in a local market (Nanning, China). Rutin standards (HPLC ≥ 98%) were purchased at Beijing Solaibao Technology Co., Ltd. (Beijing, China). Ampicillin (a800429-5g) was purchased at Shanghai Macklin Biochemical Technology Co., Ltd. Mouse IL-1β, IL-6, IL-10 and TNF-α ELISA kits were purchased from Jiangsu Meimian Industrial Co., Ltd. (Yancheng, Jiangsu, China). A nitric oxide (NO) assay kit (nitrate reductase method) was purchased from Nanjing Jiancheng Institute of Biological Engineering Co., Ltd. (Nanjing, Jiangsu, China). Absolute ethanol and sodium nitrite were purchased at Chengdu Cologne Chemical Co., Ltd. (Chengdu, China), aluminum nitrate was purchased at Tianjin Da Mao chemical reagent factory (Tianjin, China), sodium hydroxide was purchased at Chengdu Jinshan Chemical Reagent Co., Ltd. (Chengdu, China), and xylene was purchased at Tianjin Fuyu Fine Chemical Co., Ltd. (Tianjin, China). The above chemical reagents were analytical grade.

### 3.2. Animals

Kunming mice of 4–5 weeks in size, half males and females, were purchased at Tianqin Biotechnology Co., Ltd., Changsha, China. All mice were acclimatized for 1 week before the experiment (temperature 22 ± 2 °C, relative humidity 50 ± 5%, and 12 h light/dark cycle) and fed standard chow and water ad libitum [39]. All animal experimental protocols for this experiment were approved by the Institutional Animal Care and Use Committee of Guangxi University (Nanning, China) (No. Gxu-2021-145) and performed in accordance with the guide for the care and use of laboratory animals of the National Institutes of Health.

### 3.3. Determination of the Total Flavonoids

Referring to the method of standard curve preparation in a previous study [40], we weighed 10 mg of rutin standard, added an appropriate concentration of 60% ethanol solution for dissolution, and heated his solution at 55 °C to help it dissolve completely. We then added 60% ethanol solution to a 50 mL volumetric flask, at which time the concentration of rutin standard was 0.2 mg/mL. Next, we took 10 brown volumetric flasks of 10 mL, and added 0 mL, 0.2 mL, 0.4 mL, 0.6 mL, 0.8 mL, 1.0 mL, 2.0 mL, 3.0 mL, 4.0 mL, and 5.0 mL of rutin standard solution into 10 mL brown volumetric flasks. We first added an appropriate amount of 60% ethanol solution, then added 0.3 mL of 5% sodium nitrite solution. This was shaken well and left to react for 6 min. We then added 0.3 mL of 10% aluminium nitrate solution, which was shaken well and protected from light for 15 min. Next, we added 4 mL of 4% sodium hydroxide solution and added 80% ethanol solution to fix the volume to the mark. This was shaken well and left to react for 10 min. We then measured the OD value at 510 nm using a spectrophotometer. According to the corresponding OD value of the concentration of rutin standard in Figure 1, the regression equation was obtained: y = 12.626x + 0.0023, R2 = 0.9997, indicating that rutin standard has a good linear relationship at a concentration of 0–0.1 mg/mL. Furthermore, the absorbance of ATF was brought into the regression equation to obtain its concentration [41]. The yield was calculated by the following formula:(3)$$\;\;\;Yield\left(\mathrm{ATF}\right)=\left(C\times V\right)/\left(m\times 1000\right)\times 100\%$$
where *C* was the concentration of 1.00 mL filtrate of total flavonoids, *V* was the total volume of extraction solution, and *m* was the dry mass of ATF.

### 3.4. Single-Factor Experiment

Referring to the literature for the extraction of ATF [9], the purchased dried chopped pieces of Abrus cantoniensis were crushed using a grinder, passed through a 60-mesh sieve and then sealed and stored at 4 °C for future use. As one factor was investigated, all the other parameters were kept as above. The observed factors and their levels were as follows: the material-liquid ratio (1:20, 1:30, 1:40, 1:50, 1:60), ultrasonic extraction cycles (2 times, 3 times, 4 times, 5 times) using fresh and identical solvents in each repeat extraction, ethanol concentration (30%, 40%, 50%, 60%, 70%, 80%, 90%), ultrasonic time (10 min, 20 min, 30 min, 40 min, 50 min) and ultrasonic power (125 W, 150 W, 175 W, 200 W, 225 W, 250 W).

### 3.5. PBD Experiment

Based on the single-factor experiments, the factors that had a greater influence on the extraction were selected using the PBD in Design-Expert 8.0.6 software (Stat-Ease, Inc., Minneapolis, MN, USA), as shown in Table 5. Among these five factors were ethanol concentration, material-to-liquid ratio, ultrasonic time, ultrasonic extraction cycles and ultrasonic power. Using the total flavonoid extraction rate as the response value, 2 levels of high (+) and low (−) were determined for each variable, and a total of 12 experiments were conducted to determine the influence of each factor.

### 3.6. BBD Experiment

Three important influencing factors were screened based on the results of the BPD. Other influencing factors were fixed, and the experimental design was carried out using BBD in Design-Expert 8.0.6 software as shown in Table 6. Using the ATF extraction rate as the response value, response surface plots and contour plots predicted by the 3D model were used to determine the most influential variable.

### 3.7. Purification and Preparation of ATF

Through the preliminary screening of various macroporous adsorbent resins [42], the D101 macroporous adsorbent resin was found to have a strong adsorption and desorption performance on ATF. In the experiment, ATF was mainly purified by static adsorption on D101 macroporous adsorbent resin. We weighed 30 g of pre-treated D101 macroporous resin, poured it carefully into a 500 mL conical flask, added 300 mL of the ATF crude extract, put it into a shaker and shook it at 110 r/min for 4 h. We then slowly poured out the upper layer of liquid, washed the macroporous resin with distilled water, then add 400 mL of 80% ethanol solution and shook it at 110 r/min for 4 h. The ethanol solution was collected and concentrated under reduced pressure using a rotary evaporator. The ethanol was evaporated and the impurities were removed by centrifugation while hot. The supernatant was concentrated under reduced pressure again to obtain an infusion, and then freeze-dried to obtain ATF powder. We then accurately weighed 1.0 g ATF, dissolved it fully in 40% ethanol and then determined its concentration, which is the ATF purity.

### 3.8. Anti-Inflammatory Experiment

#### 3.8.1. LPS-Induced Inflammation in a Mouse Abdominal Macrophage Model

##### Preparation of Mouse Peritoneal Macrophages

Several Kunming mice were selected and injected intraperitoneally with 0.5% starch broth (1 mL/each) and killed by CO_2_ anesthesia after 2 days. The mice were immersed in 75% alcohol for 20 min, then removed and placed on sterile gauze, ventral side up. The abdominal skin of the mice was gently lifted with forceps, and 10 mL of pre-cooled PBS solution was injected intraperitoneally. The peritoneal fluid was aspirated after rubbing and pressing. The aspirated peritoneal fluid was injected into a 15 mL centrifuge tube and centrifuged at 2000 r/min for 5 min. The supernatant was discarded, and the precipitate was collected. The cells were washed with PBS, stained with Taipan Blue to confirm that the activity was above 95%, adjusted with RPMI-1640 to a concentration of 1 × 10^6^ cells/mL, added to cell culture plates and incubated at 37 °C in a 5% CO_2_ incubator.

##### Cell Activity Assay

After the cells were observed under an inverted microscope to be fully apposed, we discarded the cell supernatant. In the test group, 200 μL of ATF solution and LPS solution were prepared in RPMI-1640 at different concentrations. For blank controls, RPMI-1640 medium was added to 200 μL. Each group was set up with 4 compound holes. The cell culture plates were incubated at 37 °C in a 5% CO_2_ incubator. After 24 h, 20 μL of the prepared MTT solution was added to each well and incubated again in a CO_2_ incubator. After 4 h, the culture solution was discarded. Then, 150 μL of DMSO solution was added to each well and shaken for 10 min. Once complete dissolution of the blue-purple crystals was observed, the absorbance (optical density, OD) at 570 nm was measured by an enzyme marker.

##### Detection of NO Levels and Inflammatory Factor Secretion

After the cells were observed under an inverted microscope to be fully apposed, we discarded the cell supernatant. Blank controls were added to 200 μL of RPMI-1640 medium. The LPS model group added 200 μL of LPS solution diluted to 1 μg/mL with RPMI-1640 medium. In the ATF group, different concentrations of ATF solution and LPS solution diluted with RPMI-1640 medium were added to each well, and the final concentrations of ATF solution were guaranteed to be 10 μg/mL, 50 μg/mL and 100 μg/mL. The final concentration of LPS solution in each well was 1 μg/mL. Three replicate wells were set up for each group and incubated in a CO_2_ incubator. After 24 h, the supernatant was collected, and the NO content and the secretion of inflammatory factors in the supernatant of mouse peritoneal macrophage culture were measured according to the NO, IL-1β, IL-6, IL-10 and TNF-α ELISA kit instructions

#### 3.8.2. Acute Inflammatory Model of Xylene-Induced Ear Edema in Mice

Kunming mice, male and female, weighing 20–22 g, were randomly divided into 5 groups: the model group, positive control group (dexamethasone sodium phosphate 5 mg/kg) and dosing group (ATF 100 mg/kg, 500 mg/kg, 1000 mg/kg). Doses were administered by gavage for 5 consecutive days, once daily. A total of 1 hour after the end of the last day of administration, 0.03 mL of xylene was applied to both sides of the right ear of the mice to cause inflammation, and no drug was applied to the left ear as a comparison. After 30 min, the mice were executed, and the auricles were collected from the same location and area and weighed. The degree of oedema was assessed by the weight difference between the left and right auricles of the same animal. Auricular edema and auricular edema inhibition were calculated and the resulting data were statistically analyzed using GraphPad Prism 8 software. The weight difference in ear edema (*D*) was measured between the weight of the right and left ears of the mice. Auricular edema inhibition was defined as a percentage of the edema produced in model animals [24] using the following formula:(4)$$Inhibition=\frac{D\left(model\right)-D\left(experimental\right)}{D\left(model\right)\;}\times 100\%$$

#### 3.8.3. Cotton Pellet Granuloma Test

Kunming mice, male and female, weighing 25 ± 2 g, were randomly divided into 5 groups: the model group, positive control group (dexamethasone sodium phosphate 5 mg/kg), and dosing group (ATF 100 mg/kg, 500 mg/kg, 1000 mg/kg). Doses were administered by gavage for five consecutive days, once daily. Modeling was performed on the day of the first gavage administration, and mice were anaesthetized by intraperitoneal injection of chloral hydrate (350 mg/kg) according to their body weight. After the mice were lying still and no longer crawling, sterilized cotton balls (5 ± 0.2 mg) dried after the addition of ampicillin were transplanted under the skin of the back of the mice under aseptic conditions, and a small amount of ampicillin powder was applied to the wound after suturing to prevent wound infection. On the 6th day after completion of modeling, all mice were euthanized after CO_2_ anaesthesia, the cotton balls were removed, dried at 60 °C to a constant weight, and weighed, and the rate of granulomatous swelling inhibition was calculated.

The weight difference in granulomas (*G*) was measured between the weight of the retrieved and the implanted cotton pellet. Inhibition of cotton ball granulomas was defined as the percentage of granulomas produced in model animals, using the following formula:(5)$$Inhibition=\frac{G\left(model\right)-G\left(experimental\right)}{G\left(model\right)}\times 100\%$$

### 3.9. Statistical Analysis

Design-Expert.V8.0.6.1 (Stat-Ease Inc., Minneapolis, MN, USA) was used to calculate the coefficients of a quadratic polynomial model and to perform optimization. All data, including figures and tables, were expressed as the mean ± standard deviation (SD) from three independent experiments. Images were generated using GraphPad Prism software (La Jolla, CA, USA). In the animal experiments, mice were randomly grouped, and histological analysis was conducted in a blind manner. One-way ANOVA was used to analyse the differences between the means of normally distributed data. The results are considered statistically significant at *p* < 0.05 or *p* < 0.01.

## 4. Conclusions

The material for the study was obtained after drying, crushing and homogenizing the whole plants. In this experiment, rutin standard was used as the control for the ultrasonic-assisted solvent extraction, and the effect of different factors on the ATF extraction rate was measured. Based on the single-factor results, the PBD test was applied to screen out the three factors that had a greater influence on the ATF extraction rate, which were, in descending order, material-to-liquid ratio > ethanol concentration > number of ultrasonic extraction cycles. The best extraction process was obtained after the BBD test: the material-liquid ratio was 1:47, the number of ultrasonic extraction cycles was 4, and the concentration of ethanol was 50%. The ATF extraction rate measured under the actual conditions was 3.68%, which was not significantly different from the predicted value of 3.71%, and the relative error was calculated to be 0.81% (*n* = 3). In vitro experiments found that ATF was effective in reducing LPS-induced inflammation in mouse abdominal macrophages. In vivo experiments found that ATF significantly inhibited xylene-induced ear swelling in mice and cotton ball granuloma in mice. This suggests that ATF has a good anti-inflammatory effect. Furthermore, this study can further promote the industrial production of ATF and provide a theoretical basis for the further development of the medicinal value of Abrus cantoniensis.

## Figures and Tables

**Figure 1 molecules-27-02036-f001:**
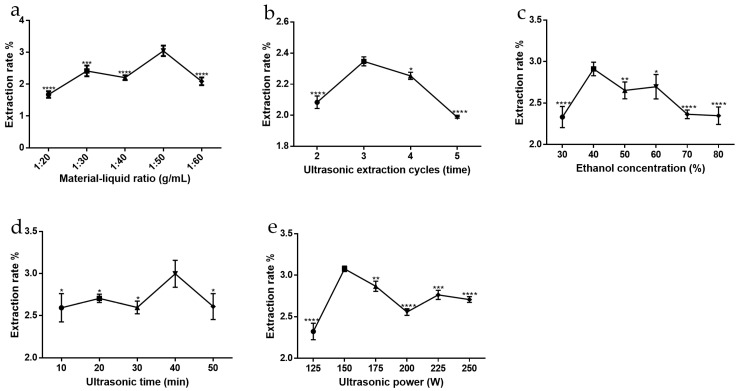
Effect of single factors on the total flavonoid extraction yield from Abrus cantoniensis. (**a**) Material-liquid ratio, (**b**) ultrasonic extraction cycles, (**c**) ethanol concentration, (**d**) ultrasonic time, (**e**) ultrasonic power. Data are expressed as mean ± SD (*n* = 3). * *p* < 0.05, ** *p* < 0.01, *** *p* < 0.001 and **** *p* < 0.0001 compared with the highest point in each group.

**Figure 2 molecules-27-02036-f002:**
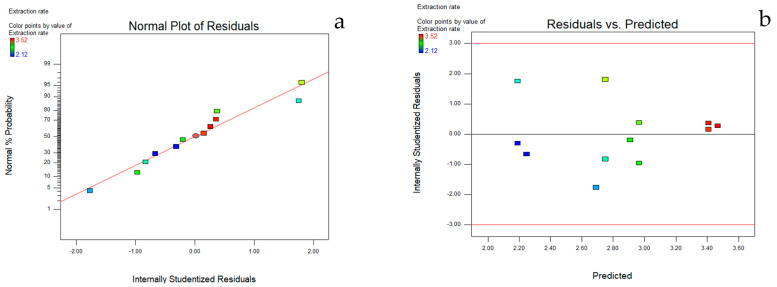
Plots of normal probability of internally studentized residuals and internally studentized residuals vs. predicted response. (**a**) The normal probability plot of the residuals. (**b**) The plots of residuals vs. the predicted responses.

**Figure 3 molecules-27-02036-f003:**
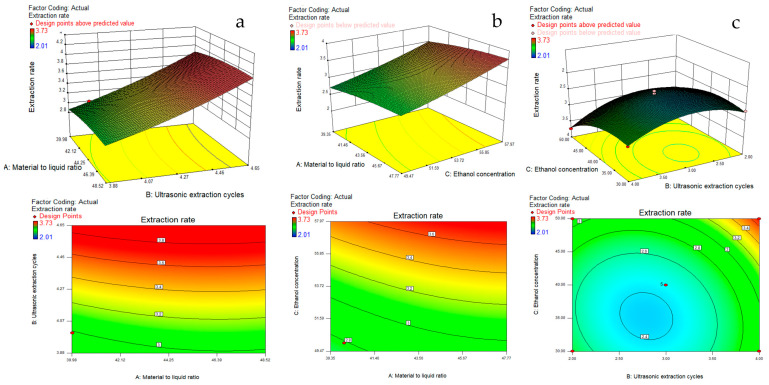
Response surface 3D plots and contour plots showing the significant interactive effect of extraction variables on the total flavonoids of Abrus cantoniensis. (**a**) Material-to-liquid ratio and ultrasonic extraction cycles, (**b**) material-to-liquid ratio and ethanol concentration, (**c**) ultrasonic extraction cycles and ethanol concentration.

**Figure 4 molecules-27-02036-f004:**
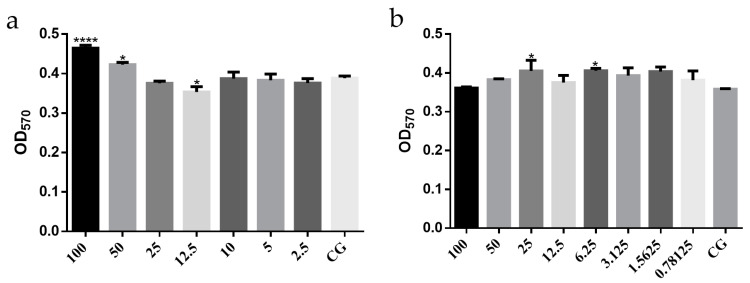
OD_570_ values of MTT results for the effect of ATF and LPS on the activity of mouse peritoneal macrophages. (**a**) The effect of ATF on the activity of mouse peritoneal macrophages. (**b**)The effect of LPS on the activity of mouse peritoneal macrophages. Data are expressed as mean ± SD. * *p* < 0.05, **** *p* < 0.0001 compared with the CG group.

**Figure 5 molecules-27-02036-f005:**
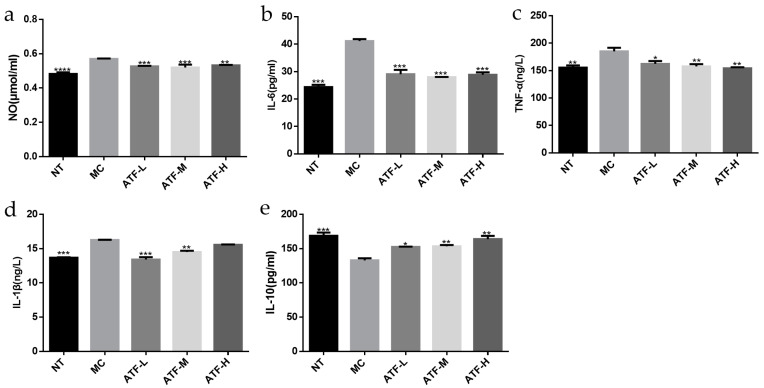
Determinations of NO content and inflammatory factor secretion in mouse peritoneal macrophages. (**a**) NO content. (**b**) IL-6 content. (**c**) TNF-α content. (**d**) IL-1β content. (**e**) IL-10 content. NT: the no-treatment control group, MC: the model control group, ATF-L: ATF 10 μg/mL, ATF-M: ATF 50 μg/mL, ATF-H: ATF 100 μg/mL. Data are expressed as mean ± SD. * *p* < 0.05, ** *p* < 0.01, *** *p* < 0.001 and **** *p* < 0.0001 compared with the MC group.

**Figure 6 molecules-27-02036-f006:**
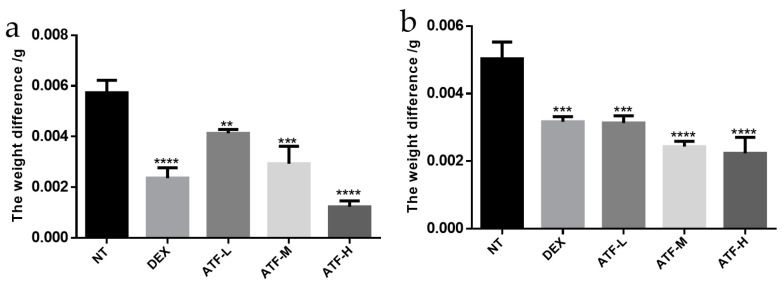
The effects of ATF on xylene-induced ear swelling and cotton ball granuloma in mice. (**a**) Inhibitory effect of ATF on xylene-induced ear swelling in mice. (**b**) Inhibitory effect of ATF on cotton ball granuloma in mice. NT: no treatment model group, DEX: dexamethasone (5 mg/kg) treatment group, ATF-L: ATF 100 mg/kg, ATF-M: ATF 500 mg/kg, ATF-H: ATF 1000 mg/kg. Data are expressed as mean ± SD. ** *p* < 0.01, *** *p* < 0.001, **** *p* < 0.0001 compared with the NT group.

**Table 1 molecules-27-02036-t001:** Plackett-Burman experimental design and response values.

Run	Material to Liquid Ratio (g/mL)	Ultrasonic Extraction Cycles (Time)	Ethanol Concentration (%)	Ultrasonic Time (Min)	Ultrasonic Power (W)	Extraction Rate (%)
1	1:60	4	30	30	125	2.53
2	1:60	2	50	50	175	3.10
3	1:60	4	50	30	125	2.35
4	1:60	4	30	50	175	2.13
5	1:60	2	30	30	175	2.12
6	1:60	2	50	50	125	2.59
7	1:40	2	30	50	125	3.04
8	1:40	2	30	30	125	2.78
9	1:40	4	30	50	175	2.87
10	1:40	4	50	30	175	3.48
11	1:40	4	50	50	125	3.44
12	1:40	2	50	30	175	3.52

**Table 2 molecules-27-02036-t002:** ANOVA for the regression quadratic model equation of PBD.

Type	Sum of Squares	df	Mean Square	F Value	Prob (P) > F
Model	2.31	3	0.77	13.69	0.0016
Material-to-liquid ratio (g/mL)	1.55	1	1.55	27.48	0.0008
Ultrasonic extraction cycles (time)	0.010	1	0.010	0.18	0.6816
Ethanol concentration (%)	0.76	1	0.76	13.40	0.0064
Residual	0.45	8	0.056		
Cor total	2.76	11			

Note: Df—the degree of freedom. F Value—the value of F, refers to the F statistic obtained by performing an F test. Prob (P) > F—the value of *p*, refers to probability and is relevant to significance.

**Table 3 molecules-27-02036-t003:** Box-Behnken experimental design and response values.

Run	Material-to-Liquid Ratio (g/mL)	Ultrasonic Extraction Cycles (Time)	Ethanol Concentration (%)	Predicted Extraction Rate (%)	Actual Extraction Rate (%)
1	0	−1	−1	2.73	2.70
2	0	−1	1	3.09	3.07
3	0	1	−1	2.97	3.00
4	0	1	1	3.69	3.73
5	1	−1	0	2.28	2.27
6	1	1	0	2.60	2.54
7	1	0	1	2.61	2.62
8	1	0	−1	1.97	2.01
9	−1	0	−1	2.35	2.34
10	−1	−1	0	2.46	2.51
11	−1	1	0	2.98	2.99
12	−1	0	1	2.79	2.75
13	0	0	0	2.45	2.41
14	0	0	0	2.45	2.63
15	0	0	0	2.45	2.33
16	0	0	0	2.45	2.56
17	0	0	0	2.45	2.33

**Table 4 molecules-27-02036-t004:** ANOVA for the regression quadratic model equation of BBD.

Type	Sum of Squares	df	Mean Square	F Value	Prob (P) > F
Model	2.46	9	0.27	21.75	0.0003
A	0.17	1	0.17	13.18	0.0084
B	0.37	1	0.37	29.14	0.0010
C	0.56	1	0.56	44.79	0.0003
AB	0.011	1	0.011	0.88	0.3797
AC	0.010	1	0.010	0.80	0.4016
BC	0.032	1	0.032	2.58	0.1521
A^2^	0.34	1	0.34	27.22	0.0012
B^2^	0.71	1	0.71	56.50	0.0001
C^2^	0.29	1	0.29	23.17	0.0019
Residual	0.088	7	0.013		
Lack of Fit	0.013	3	0.004308	0.23	0.8713
Pure Error	0.075	4	0.019		
Cor Total	2.54	16			

Note: A—material-to-liquid ratio (g/mL), B—ethanol concentration (%), C—ultrasonic extraction cycles (time). Df—the degree of freedom. F Value—the value of F, refers to the F statistic obtained by performing an F test. Prob (P) > F—the value of *p*, refers to probability and is relevant to significance.

**Table 5 molecules-27-02036-t005:** Plackett-Burman design factor level.

Factor	Extraction Conditions	Low (−)	High (+)
X_1_	Material to liquid ratio (g/mL)	1:40	1:60
X_2_	Ultrasonic extraction cycles (time)	2	4
X_3_	Ethanol concentration (%)	30	50
X_4_	Ultrasonic time (min)	30	50
X_5_	Ultrasonic power (W)	125	175

**Table 6 molecules-27-02036-t006:** Box-Behnken design factor level.

Level	Factor		
	Material to Liquid Ratio (g/mL)	Ultrasonic Extraction Cycles (Time)	Ethanol Concentration (%)
−1	1:40	2	30
0	1:50	3	40
1	1:60	4	50

## Data Availability

All available data are contained within the article.

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
