# Peer review of "Optimization of Ultrasonic-Assisted Extraction of Total Flavonoids from Abrus Cantoniensis (*Abriherba*) by Response Surface Methodology and Evaluation of Its Anti-Inflammatory Effect"

_molecules, 2022, doi:10.3390/molecules27072036_

Round 1

Reviewer 1 Report

Article
Optimization of ultrasonic assisted extraction of total flavonoids from Abrus cantoniensis (Abriherba) by response surface methodology and its anti-inflammatory effect

A brief summary
The article is noteworthy but needs a lot of corrections and clarifications. Improvements to the keywords and text are also necessary - all the proposed words are used in the title.

Broad comments
1.    Why is the order of extraction factors different in different parts of the article? For example, in the summary, line 94, and table 5. In how many replicates were these studies conducted? The results of the statistical analysis could be added to Figure 1, to be able to draw the conclusions set out in section 2.1. .
2.    The term ‘liquid material ratio’ (lines 21, 22, 93, 175, 435, 443) is totally inappropriate and misleading in the context of solid-liquid extraction. A better form is found in line 74 (liquid-material ratio), in lines 62, 94, ... , 542, 544 (liquid to material ratio) and in line 164 (liquid-to-material ratio). In lines 423 and 201 the order is reversed (material-liquid ratio) and this form is the most appropriate. It is traditionally written/expressed as e.g.: 1:10, w/v (mass per volume). Interestingly, such a notation appears in Table 1 (but with the wrong heading mL/g). Please get it right.
3.    The term 'Ultrasonic extraction times (time)' is misleading, especially when compared with 'Ultrasonic time (min)'. It refers to multiple periodic extraction. In addition, it should be specified whether repeated extractions were carried out with the same or a new (clean) portion of solvent (line 424). Why was not a single cycle extraction tested? 
4.    The Authors should also insert the correct designation of degrees Celcius and insert spaces and commas in the correct places (e.g. line 420, 425).

Specific comments
Line 390. “All mice 120 were acclimatized ...” ?

Lines 24, 426, Table 1. The watt (symbol: W, not w) is a unit of power in the International System of Units (SI).

Line 416. There is a lowercase ‘m’ in the formula and an uppercase ‘M’ in the description.

Line 538. It seems that instead of a sentence ‘The whole plant powder was used as the raw material’ the following sentence is more appropriate ‘The whole plant was tested after drying, crushing and homogenisation.’ or ‘The material for the study was obtained after drying, crushing and homogenising the whole plants.’

Table 1, 3, 5 and 6, Figure 2. What is the unit of ‘time’? As it refers to multiple periodic extraction, the more appropriate term here is extraction cycles, as in lines 98, 172, 185.  

Author Response

Dear Reviewer:

  We are grateful for your valuable suggestions, which have been of great benefit to us and have left us with a sincere admiration for your high level of professionalism and presentation.Based on your suggestions, we have discussed and revised the full text, please see the attachment for your specific response.

Yours sincerely,

All authors of molecules-1627733

Reviewer 2 Report

The manuscript submitted by Wu et al. reported on the “Optimization of ultrasonic assisted extraction of total flavonoids from Abrus cantoniensis (Abriherba) by response surface methodology and its anti-inflammatory effect”. I suggest the publication, after minor revision, of the article in “Molecules” as the research is detailed; the approach, the results and conclusion are well supported by experimental data and interesting for the scientific audience.

I suggest just some correction:

  • Is my opinion, the title should be modified as below suggested:

Optimization of ultrasonic assisted extraction of total flavonoids from Abrus cantoniensis (Abriherba) by response surface methodology and evaluation of anti-inflammatory effect

  • There are some repetition in the text: e.g. row 60 and row 72 reported the same sentence. Row 60: a single factor test and orthogonal test were used to optimize 60 the ATF ultrasonic extraction process; row 72: Single factor test and orthogonal test were used to optimize the ATF ultra-72 sonic extraction process. Please, read carefully the text.
  • The extraction temperature is a significant factor influencing the extraction yield and this parameter was omitted. Please. Add it.
  • Are you able to explain why you find no effect on the extraction yield by changing sonication time and sonication frequencies as many other articles states the contrary? Read this article as an example     https://doi.org/10.3390/antiox8100425
  • the effect of the independent variable on the total flavonoid yield could be plotted as diagram, es total… vs liquid/solid ratio

Author Response

Dear Reviewer:

  We appreciate your valuable suggestions, which have been of great benefit to us and have given us a genuine sense of admiration for your high level of professionalism and presentation.The full text has been discussed and revised in the light of your suggestions, please see the attachment for your specific response.

Yours sincerely,

All authors of molecules-1627733

Reviewer 3 Report

The manuscript "Optimization of ultrasonic assisted extraction of total flavonoids from Abrus cantoniensis (Abriherba) by response surface methodology and its anti-inflammatory effect" is devoted to extraction of flavonoids from Abriherba using ethanol-water mixtures and in-vitro and in-vivo study of their anti-inflammatory effects. Conditions of ultrasound-assisted extraction, such as extraction time, liquid-to-solid ratio, US-power etc. Only one characteristic of extracts (total flavonoid content) was used for optimization.

Generally, the study is quite interesting, the manuscript is written clearly, well structured, but some missing in describing of the experiment may be found in the manuscript. I think the manuscript may be published in the Molecules journal after minor revision after taking into account some of the remarks described below

  1. Section “2.1.2.Influence of ultrasonic extraction times”: The authors select not optimal extraction conditions. It would be better to select time, liquid-to-material ratio, ethanol concentration and US-power, before studying extraction times effect. This point should be clarified in the text.
  2. Which extract (obtained in which conditions) was used in the anti-inflammatory effects? How anti-inflammatory effects relate with the flavonoid content in extract? Did you tested the extracts obtained at various condition?
  3. Which temperature of extraction did you use? It is highly important to mention and discuss it in this work.

Author Response

Dear Reviewer:

  We are honoured and pleased to receive your valuable suggestions, which have been of great benefit to us. In all fairness, this has left us with a genuine sense of admiration for your high level of professionalism and presentation.Following your suggestions, we have discussed and revised the full text and have provided a detailed explanation of your questions, please see the attachment for details.

Yours sincerely,

All authors of molecules-1627733
